# Influence of rice varieties, organic manure and water management on greenhouse gas emissions from paddy rice soils

Ei Phyu Win[ID][1]*, Kyaw Kyaw Win[1], Sonoko D. Bellingrath-Kimura[2], Aung Zaw Oo[3¤]

1 Department of Agronomy, Yezin Agricultural University, Yezin, Myanmar, 2 Institute of Land Use Systems, Faculty of Life Science, Humboldt-Universität zu Berlin, Berlin, Germany, 3 Institute for Agro-Environmental Science, National Agriculture and Food Research Organization, Tsukuba, Ibaraki, Japan

¤ Current address: Japan International Research Center for Agricultural Sciences, Tsukuba, Ibaraki, Japan
* eiphyuwin@yau.edu.mm

**Data Availability Statement:** All relevant data are within the paper.

**Funding:** This study was supported by the Cuomo Foundation (https://www.cuomo.foundation/). The

## Abstract

The study is focused on impact of manure application, rice varieties and water management on greenhouse gas (GHG) emissions from paddy rice soil in pot experiment. The objectives of this study were a) to assess the effect of different types of manure amendments and rice varieties on greenhouse gas emissions and b) to determine the optimum manure application rate to increase rice yield while mitigating GHG emissions under alternate wetting and drying irrigation in paddy rice production. The first pot experiment was conducted at the Department of Agronomy, Yezin Agricultural University, Myanmar, in the wet season from June to October 2016. Two different organic manures (compost and cow dung) and control (no manure), and two rice varieties; Manawthukha (135 days) and IR-50 (115 days), were tested. The results showed that cumulative $CH_4$ emission from Manawthukha (1.084 g $CH_4$ $kg^{-1}$ soil) was significantly higher than that from IR-50 (0.683 g $CH_4$ $kg^{-1}$ soil) (P<0.0046) with yield increase (P<0.0164) because of the longer growth duration of the former. In contrast, higher cumulative nitrous oxide emissions were found for IR-50 (2.644 mg $N_2O$ $kg^{-1}$ soil) than for Manawthukha (2.585 mg $N_2O$ $kg^{-1}$ soil). However, IR-50 showed less global warming potential (GWP) than Manawthukha (P<0.0050). Although not significant, the numerically lowest $CH_4$ and $N_2O$ emissions were observed in the cow dung manure treatment (0.808 g $CH_4$ $kg^{-1}$ soil, 2.135 mg $N_2O$ $kg^{-1}$ soil) compared to those of the control and compost. To determine the effect of water management and organic manures on greenhouse gas emissions, second pot experiments were conducted in Madaya township during the dry and wet seasons from February to October 2017. Two water management practices {continuous flooding (CF) and alternate wetting and drying (AWD)} and four cow dung manure rates {(1) 0 (2) 2.5 t $ha^{-1}$ (3) 5 t $ha^{-1}$ (4) 7.5 t $ha^{-1}$} were tested. The different cow dung manure rates did not significantly affect grain yield or greenhouse gas emissions in this experiment. Across the manure treatments, AWD irrigation significantly reduced $CH_4$ emissions by 70% during the dry season and 66% during the wet season. Although a relative increase in $N_2O$ emissions under AWD was observed in both rice seasons, the global warming potential was significantly reduced in AWD compared to CF in both seasons (P<0.0002, P<0.0000) according to reduced emission in $CH_4$. Therefore, AWD is the

funders had no role in study design, data collection and analysis, decision to publish, or preparation of the manuscript.

**Competing interests:** The authors have declared that no competing interests exit.

effective mitigation practice for reducing GWP without compromising rice yield while manure amendment had no significant effect on GHG emission from paddy rice field. Besides, AWD saved water about 10% in dry season and 19% in wet season.

## Introduction

Developing new strategies is necessary to achieve the dual goals of ensuring food security and protecting natural resources and the environment through reduced greenhouse gas (GHG) emissions [1, 2]. It is estimated that nitrous oxide ($N_2O$) and methane ($CH_4$) emissions may increase by 35–60% and 60%, respectively, by 2030 [3]. Flooded rice soils are an important source of global $CH_4$ emissions [4, 5], and rice-based cropping systems can emit substantial amounts of $N_2O$ [6] during the rice season itself [7].

Rice is one of the most important cereal grains and staple food crops globally and is particularly important in Asia [8]. Rice paddies contribute to the emission of the two most important GHGs; methane and nitrous oxide. IPCC [9] reported that rice fields contribute about 30% and 11% of global agricultural $CH_4$ and $N_2O$ emissions, respectively. With a linearly increasing rate of 0.26% per year during the recent few decades, the atmospheric $N_2O$ concentration has increased by 18% compared to the preindustrial level. Methane and nitrous oxide have long atmospheric lifetimes of 12 and 114 years, respectively, and account for 20% and 7%, respectively, of global radiative forcing [10]. The high global warming potential (GWP) of $CH_4$ and $N_2O$, 34 and 298 times that of $CO_2$ at a 100-year time horizon, makes them major contributors to climate change [11]. In recent years, suitable management practices have been developed for achieving both improvement in rice yields and mitigation of GHG emissions, which include the development of new rice varieties [12], the application of manure such as cow dung [13], the selection of appropriate cultivation methods [14] and the timing of drainage [15].

The magnitude of $CH_4$ emissions from rice plants is regulated by complex and dynamic interactions among the plants, environment, and microorganisms [16]. Methane produced in flooded rice soils is emitted to the atmosphere by molecular diffusion, ebullition or plant-mediated transport. Approximately 80–90% of the total $CH_4$ flux is emitted to the atmosphere from the rhizosphere via the rice plant [17]. An increase in plant biomass [18] and tiller number [19] enhanced $CH_4$ oxidization activity by enlarging the volume of aerenchyma and enhancing $O_2$ transport from the atmosphere to the rhizosphere. Ma et al. [20] revealed that a hybrid rice variety with 50–60% higher shoot biomass emitted less $CH_4$ than an indica rice variety, possibly due to higher $CH_4$ oxidization activity.

Nitrous oxide is produced as a by-product of nitrification, denitrification, nitrifier denitrification, etc. and moisture content is a key factor governing $N_2O$ production in soils [21]. Ciarlo et al. [22] found that denitrification is dominant pathway for $N_2O$ emission when the water-filled pore space in soils is high (80%), and if the soils were saturated than this level, most of $N_2O$ would be reduced to nitrogen.

Selecting a rice variety that has high productivity and low GHG emissions is crucial for improving crop yield and mitigating climate change; however, research examining the effects of rice varieties has mostly focused on $CH_4$ flux so far [12, 23, 24], with little focus on $N_2O$ flux [25]. Many studies reported that the effect of rice varieties on methane emissions is mostly related to rice growth performance, i.e., the number of plant tillers and above- and below-ground biomass, root exudates and root arenchyma [26–30]. Although significant positive

relationship have been found between rice biomass and methane fluxes [31, 32], a comparison of rice varieties has produced different results [33, 34].

Organic residue amendments have been practised to improve soil fertility in paddy production. The organic matter in paddy fields originates from both direct by-products of rice production (such as sloughed-off root cells and root exudates) and added materials (manures and previous crop residues). The addition of organic carbon to the soil, whether it comes from the disposal of crop residues or as organic fertilizer, appears to be the most important factor in methane production [32]. Waterlogged conditions are ideal for the decomposition of organic matter in paddy fields. The methane production from rice soil can be increased by addition of cow manure as a source of organic material [35]. Nitrous oxide emissions from applied fertilizer and manures can vary with different environmental factors (e.g., climate and soil conditions), crop factors (e.g., crop type and crop residues), and management practices (e.g., type of manure and fertilizer, application rate, time of application) [36].

Although there are wide range of factors influencing methane emission from paddy soil [37–39], water management and organic amendment are the two main drivers of methane production. Under anaerobic soil conditions, methanogens (methane-producing bacteria) produce methane by oxidation of organic matter during anaerobic respiration. Flooded rice soils are known to have strong denitrification activity emitting some amount of $N_2O$ from rice soils. However, it is also guided by the water management conditions of the rice field. Nitrate and nitrite in rice soils is limited due to submerged conditions. The oxygen supply due to decomposition of organic matter, roots, and also through vascular transport via tillers may help in production of nitrate in rice soils. Sometimes due to prolonged submergence of the rice fields, the soil nitrate and nitrite N (available due to mineralization of organic matter) is completely reduced to $N_2$ gas thereby resulting in low $N_2O$ emission. This varies with the prevailing conditions of the rice field. Although minimal $N_2O$ emissions are likely from flooded soils, some off-site (indirect) $N_2O$ emissions are likely from irrigated rice production due to the addition of nitrogen fertilizer to fields [40].

Myanmar ranks the sixth largest production for rice in the world. Rice is the country's most important crop and is grown on 7.3 million ha [41]. The conventional rice production method commonly used by the farmers in Myanmar includes transplanting old seedlings (30–45) under continuous flooding conditions and the intensive use of organic fertilizers such as manure or compost. However, there is very limited information on methane emissions from the flooded rice fields of Myanmar, although more than half of the cultivated area is under to rice production [13, 42]. Thus, the objectives of this study were a) to assess the effect of different types of manure amendments and rice varieties on greenhouse gas emissions and b) to determine the optimum manure application rate to increase rice yield while mitigating GHG emissions under alternate wetting and drying irrigation in paddy rice production.

## Materials and methods

The first pot experiment was conducted at open field, Department of Agronomy, Yezin Agricultural University (19° 45'N and 96° 6'E), Myanmar, during the wet season (June–October), 2016 to study the local production potential of GHG emission in this area. A two-factor factorial experiment with completely randomized design was used with 3 replications. The factor A was assigned into two categories of organic manure (compost and cow dung) and control (no manure). The compost is collected from straw compost making process and stored for ten months. The cow dung is resulted from farmer traditional heap method for ten months. The amount of organic manure for cow dung and compost treatments was based on the nitrogen content of the organic manure analysis. The recommended chemical fertilizer amounts are 60

kg N ha$^{-1}$, 30 kg P$_2$O$_5$ ha$^{-1}$, and 20 kg K$_2$O ha$^{-1}$ for all treatments. The recommended amounts of nitrogen fertilizer of compost and cow dung treatments were replaced by compost and cow dung manures which was based on same amount of carbon content. Therefore, 3 t ha$^{-1}$ of compost was applied to compost treatment, and 3.15 ton cow dung + chemical fertilizer (22.15 kg N ha$^{-1}$) was applied to cow dung treatment to get the recommended amount of nitrogen fertilizer based on their nitrogen analysis. The control treatment received only recommended amount of chemical nitrogen fertilizer. The factor B was two types of rice varieties: Manawthukha (135 days) and IR-50 (115 days). The two rice varieties were grown in a concrete pot (52.5 cm in diameter and 45 cm in height). The soil was collected from a lowland irrigated rice field. Twenty-one-day-old seedlings were transplanted with two seedlings per pot. Compost and cow dung manures were broadcasted at 14 days before transplanting to avoid transplanting shock due to the manure decomposition process. The soil was analysed for pH (1:5 soil: water suspension), electrical conductivity (1:5 soil: water suspension), total N% (Kjaldehl distillation method), organic matter% (Tyurin's method), calcium chloride extractable SO$_4$-S (Turbidity method) and Texture (pipette method). The manures were analysed for total N% (Kjaldehl distillation method), total P% (Molybivanado phosphoric acid method), total K% (Wet digestion with HNO3: HCLO4 (4:1), total S% (Turbidity method) and organic carbon % (Tyurin's method). Table 1 shows the physiochemical properties of the soil and manures. The recommended amount of T-super (30 kg P$_2$O$_5$ ha$^{-1}$) was applied as basal fertilizer, and the recommended amount of potash (20 kg K$_2$O ha$^{-1}$) was applied with two split applications (as basal fertilizer and at the panicle initiation stage) to all treated pots. The water level was maintained at 5 cm throughout the rice growing period except during the drying period before harvest. During the rice growing season, weather data were recorded at the Department of Agronomy and are shown in Fig 1. The average minimum and maximum temperatures during the rice growing season (wet season) were 23.9 and 31.8˚C, respectively, with 739 mm of rainfall.

The second pot experiment was conducted in a farmer's field, Madaya Township (22˚ 13' 0" N and 96˚ 7' 0" E), Myanmar, during the dry and wet seasons (February–October 2017) to assess the effect of cow dung manure and water management on greenhouse gases emissions from paddy rice soils. The pots were arranged in a two-factor factorial experiment with completely randomized design with three replications. Water management (continuous flooding (CF) and alternate wetting and drying (AWD) was arranged as factor A. Different rates of organic manure were assigned as factor B. In this study, cow dung manure was applied as an organic source based on the reduced global warming potential (GWP) value in previous study and widely used in the study area. The cow dung manure treatments (OM$_0$ = no cow dung, OM$_1$ = half of the recommended cow dung (2.5 t ha$^{-1}$), OM$_2$ = the recommended rate of cow dung (5 t ha$^{-1}$) and OM$_3$ = one and a half times the recommended rate of cow dung (7.5 t ha$^{-1}$), were applied seven days before transplanting. The recommended rate of cow dung manure is 5 t ha$^{-1}$. Each pot received the recommended fertilizer at the rates of 90 kg N ha$^{-1}$, 30 kg P$_2$O$_5$ ha$^{-1}$, and 20 kg K$_2$O ha$^{-1}$. Urea, T-super and potash were used as nutrient sources. Urea was applied as three equal split applications at the active tillering, panicle initiation and heading growth

**Table 1. Physiochemical properties of experimental soil and organic manures used in the first pot experiment.**

| Item | Total N% | Total P% | Total K% | O.C% | | | |
|---|---|---|---|---|---|---|---|
| Cow dung | 1.2 | 1.0 | 2.1 | 23.3 | | | |
| Compost | 2.0 | 2.9 | 1.9 | 24.5 | | | |
| Item | pH | EC (dS/m) | Total N% | OM% | SO4-S (mg/kg) | Texture % | | |
| | | | | | | Sand | Silt | Clay |
| Soil | 6.7 | 0.4 | 0.2 | 2.7 | 12 | 79.74 | 13.28 | 6.98 |

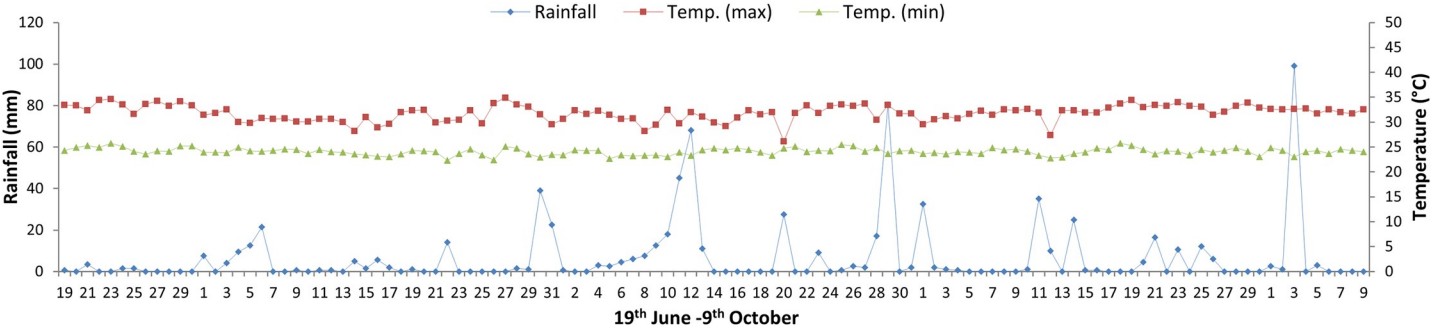

**Fig 1. Daily rainfall, maximum and minimum temperatures in Yezin Agricultural University, Myanmar during wet season, 2016.**

stages. T-super was applied only as a basal fertilizer at one day before transplanting, and potash fertilizer was applied in two equal split applications as a basal fertilizer and at panicle initiation. The soil was collected from a lowland irrigated rice field and analysed for pH (1:5 soil: water suspension), available N (Alkaline permanganate method), available P (9C-Olsen's P-Malachite green), available K (1N Ammonium acetate extraction), total N% (Kjaldehl distillation method), organic matter% (Tyurin's method), cation exchange capacity (CEC) (Leaching method) and texture (Pipette method). The cow dung manure was collected from farmer traditional heap method stored for ten months, and analysed for total N% (Kjaldehl distillation method) and organic carbon % (Tyurin's method). Table 2 shows the physiochemical properties of the soil and cow dung.

IR-50 rice variety (115 days) was used based on reduced emission in previous study and widely grown in the study area. Dry-season rice was transplanted on 1st February 2017 and harvested on 14 May 2017. Wet-season rice was transplanted on 8 July 2017 and harvested on 11st October 2017. Just after transplanting, a base (40 cm in diameter with 2.5 cm water seal, 5 cm in height) was placed around the plants used for gas sampling to avoid disturbing the environmental conditions around the rice plants during chamber deployment in both experiments. The base was equipped with a water seal to ensure a gas-tight closure. The base remained embedded in the soil throughout the rice growing period. Water tubes (PVC pipe-

**Table 2. Physiochemical properties of experimental soil and cow dung manure used in the second pot experiment.**

| Analytical Item | Unit | Analytical Result | |
|---|---|---|---|
| Soil pH | | 7.4 | Moderately alkaline |
| Available N | mg kg$^{-1}$ | 50 | Low |
| Available P | mg kg$^{-1}$ | 13 | Medium |
| Available K | mg kg$^{-1}$ | 78 | Low |
| Total N | % | 0.17 | |
| Organic matter | % | 1.8 | Low |
| CEC | cmol$_c$ kg$^{-1}$ | 11 | Low |
| Sand | % | 87 | |
| Silt | % | 4 | |
| Clay | % | 9 | |
| Textural class | | | Loamy sand |
| **Cow dung manure** | | **Dry season** | **Wet season** |
| Total N | % | 1.32 | 1.2 |
| Organic carbon | % | 16 | 23.3 |

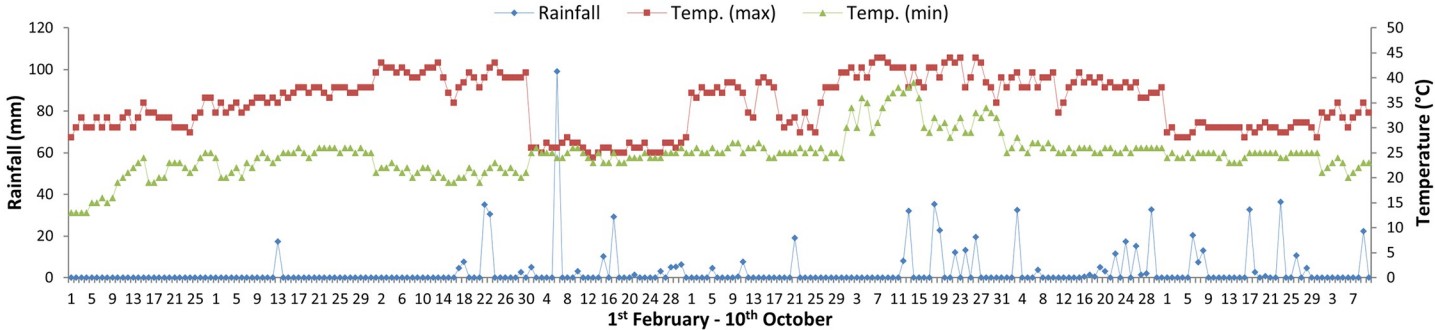

**Fig 2. Daily rainfall, maximum and minimum temperatures in Madaya township, Myanmar during dry and wet seasons, 2017.**

25 cm height with six row holes each 2.5 cm apart) were installed in the AWD pots at a depth of 15 cm below the soil surface between the seedlings and the base just after transplanting. For AWD pots, whenever there was no water in the water tube, irrigation water was applied to a 5 cm depth above the soil surface. The irrigation interval ranged from 4 to 9 days, and the amount ranged from 7 to 13 litres depending on the different rates of cow dung manure in the AWD pots. Withdrawal of water was started one week before the harvest period in all irrigated pots. During the rice growing seasons, weather data were collected from the Department of Agricultural Research, Madaya and are shown in Fig 2. The average minimum and maximum temperatures were 21.8˚C and 35.6˚C during the dry season and 26.8˚C and 35.7˚C during the wet season, respectively. The total rainfall amounts were 201.7 mm during the dry season and 420.6 mm during the wet season.

## Gas sample collection, analysis and calculation

A two-bonded chamber with a total capacity of 77 L (93 cm height) was used for collecting the gas sample. To thoroughly mix the gases in the chamber, the chamber was equipped with a small 12 volt DC fan connected with three 9-volt dry battery [43]. For $CH_4$ calculation, the temperature was recorded with a digital thermometer (TT-508 Tanita, Tokyo, Japan). To compensate for the air pressure changes between the increased temperature and gas sampling, an air buffer bag (1-L Tedlar bag) was attached to the chamber. The silicon rubber tube connected with three-way stop cock was inserted airtight into a hole on the chamber. The gas sample was taken with an airtight 50 ml syringe by connecting it to the three-way stop cock and then transferred to a 20 ml pre-evacuated glass vial.

Gas sampling was performed at 7-day intervals starting from 1 day after transplanting until harvest by the closed chamber method in pot experiment 1. In pot experiment 2, the first two gas samplings were performed at one-week intervals after transplanting, and later gas samplings were performed at 10-day intervals. The gas samples were collected from 9:00 am until 12:00 am and three times (0, 15, 30 min) for each treatment for gas flux calculation.

Methane and $N_2O$ concentrations were analysed with a gas chromatograph (GC 2014, Shimadzu Corporation, Kyoto, Japan) equipped with a flame ionization detector (FID) and an electron capture detector (ECD). The amount of $CH_4$ and $N_2O$ fluxes was calculated by using the following equation:

$$Q = (V/A) \text{ x } (\Delta c/\Delta t) \text{ x } (M/22.4) \text{ x } (273/K)$$

where   Q = the flux of gas (mg m$^{-2}$ min$^{-1}$)
         V = the volume of the chamber (m$^3$)

A = the base area of the chamber ($m^2$)

($\Delta c/\Delta t$) = the rate of increase or decrease in the gas concentration (mg $m^{-3}$) per unit time

(min)

M = the molar weight of the gas

K = Kelvin temperature of the air temperature inside the chamber

Total emissions were calculated by interpolation method of sample gas analysis at each gas measurement for the growing period. In this study, the IPCC factors were used to calculate the combined GWPs for 100 years (GWP = ($25 \times CH_4$) + ($298 \times N_2O$)) in kg $CO_2$-equivalents $ha^{-1}$ [9] for $CH_4$ and $N_2O$. The yield was recorded from each pot. The grains were threshed, cleaned and sun-dried. Yields were adjusted at 14% moisture by using the following formula to remove the error due to the different moisture content, and grain moistures were measured by using a grain moisture metre (model: GMK-303RS).

Adjusted grain weight at 14% moisture level = A x W

where    A = Adjustment coefficient

W = Weight of harvested grains

$$A = \frac{100 - \text{moisture}\%}{86}$$

In both water management practices, the water was applied with 1.2 L water cup. The total amount of water applied throughout the growing season was recorded and water saving in AWD was calculated.

The greenhouse gas intensity (GHGI) was calculated by dividing the GWP by the rice grain yield [44–46].

$$GHGI = GWP/\text{grain yield (kg } CO_2-\text{eq. kg}^{-1} \text{ grain)}$$

## Statistical analysis

The data were analysed by using Statistix (version 8.0). Mean comparisons were performed by least significant difference (LSD) test at the 5% level.

## Results

### First pot experiment during the wet season, 2016

**Methane emission.**   During the early growth stage, low $CH_4$ emission was observed for both varieties until 36 days after transplanting (DAT), and then $CH_4$ emission flux gradually increased with some fluctuations until harvest (Fig 3). The mean cumulative $CH_4$ emissions from the control, compost, and cow dung treatments were 0.893, 0.951 and 0.808 g $CH_4$ $kg^{-1}$ soil, respectively. Although the effect was not significant, cow dung amendment reduced cumulative $CH_4$ emissions by 9.5% compared with the control and 15% compared to compost. When comparing the two rice varieties, the total cumulative $CH_4$ emissions were significantly higher in the Manawthukha variety (1.084 g $CH_4$ $kg^{-1}$ soil) than in the IR-50 variety (0.683 g $CH_4$ $kg^{-1}$ soil) (P<0.0046) (Table 3). The IR-50 variety reduced cumulative $CH_4$ emissions by 37% compared to the Manawthukha variety. During the rice growing season, cumulative $CH_4$ emissions were higher in later growth stages (reproductive and ripening) than in the vegetative growth stage.

**Nitrous oxide emission.**   High nitrous oxide emission was observed during very early growth until 15 DAT in both varieties. After that, a small amount of $N_2O$ emission was found until harvest (Fig 4). Although there was no significant difference in $N_2O$ emissions among the

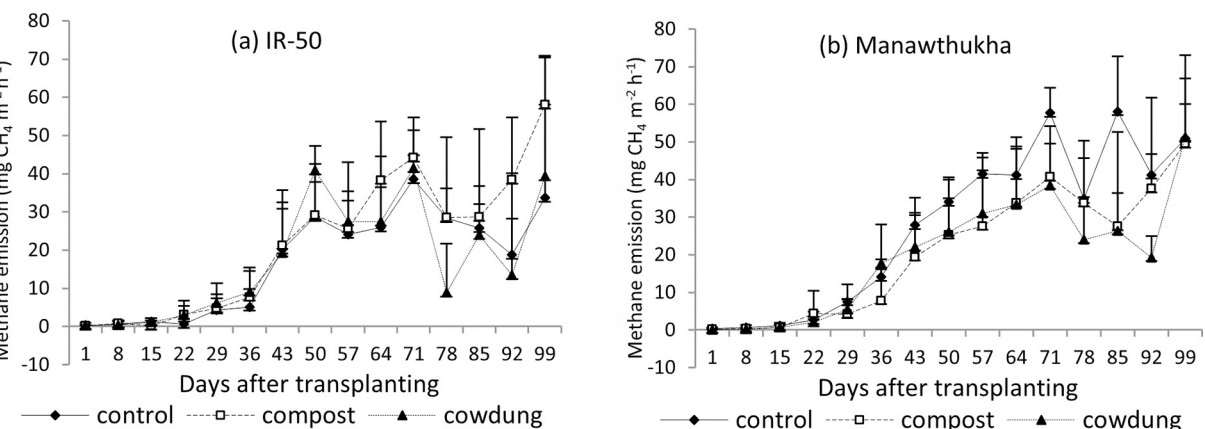

**Fig 3. Methane emission of rice varieties at Yezin Agricultural University during the wet season, 2016.** Mean value±standard deviation (n = 3).

manure treatments, higher emission (3.218 mg $N_2O$ $kg^{-1}$ soil) was recorded from the control (no manure) compared to the compost (2.491 mg $N_2O$ $kg^{-1}$ soil) and cow dung (2.135 mg $N_2O$ $kg^{-1}$ soil) (Table 1). Cow dung manure reduced cumulative $N_2O$ emissions by 33.7% compared with the control and 14.3% compared to compost. There was also no significant difference in cumulative $N_2O$ emissions among the tested varieties: Manawthukha variety (2.585 mg $N_2O$ $kg^{-1}$ soil), IR-50 (2.644 mg $N_2O$ $kg^{-1}$ soil) (Table 3).

**Global warming potential (GWP).** GWP was not significantly different among manure treatments, although higher GWP (17.2 Mt $CO_2$-eq. $ha^{-1}$) was observed in the compost treatment followed by the control (16.3 Mt $CO_2$-eq. $ha^{-1}$) and cow dung treatment (14.6 Mt $CO_2$-eq. $ha^{-1}$) (Fig 5A). There was a significant difference in GWP among the varieties (P<0.0050). A higher GWP was observed for Manawthukha (19.5 Mt $CO_2$-eq.$ha^{-1}$) than for IR-50 (12.5 Mt $CO_2$-eq. $ha^{-1}$).

**Rice yield.** Grain yield was not significantly affected by the manure treatments. However, the numerically highest grain yield (122.3±2.4 g $plant^{-1}$) was recorded from the control (no manure), followed by the compost (102.6±13.6 g $plant^{-1}$) and cow dung treatments (79.6±15.6 g $plant^{-1}$). There was significant different in grain yield between the varieties Manawthukha (121.5±15.0 g $plant^{-1}$) and IR-50 (81.6±8.0 g $plant^{-1}$) (P<0.0164) (Table 3).

**Greenhouse gas intensity.** Greenhouse gas intensity was not affected by manure management and varieties (Table 3). However, across manure management treatments, higher GHGI values were found in Manawthukha (3.8 kg $CO_2$-eq. $kg^{-1}$ grain) than in IR-50 (3.5 kg $CO_2$-eq. $kg^{-1}$ grain) (Fig 5B). There was no interaction between manure and rice varieties on the GHGI.

## Second pot experiment during the dry and wet seasons, 2017

**Methane emission.** The seasonal methane emissions of rice are shown in Figs 6 (dry season) and 7 (wet season). In the dry season, high methane emissions were observed in the early growth stage and then decreased until harvest under both water regimes. In the wet season, a slight increase in emissions was recorded in the early growth stage, emissions peaked in the middle stage, and gradually decreased until harvest. There were significant differences in cumulative $CH_4$ emissions among water management practices (P<0.0003) (Table 4). Higher cumulative $CH_4$ emissions were observed under CF than AWD. Despite no significant difference among the cow dung manure rates, a generally higher amount of cow dung manure produced more methane emissions than lower rates (Fig 12A).

**Table 3. Effects of manure and rice variety on greenhouse gases emission and grain yield of rice during wet season, 2016.**

| Treatment | $CH_4$ (g $kg^{-1}$ soil) | $N_2O$ (mg $kg^{-1}$ soil) | Grain yield (g $plant^{-1}$) | GHGI (kg $CO_2$-eqv $kg^{-1}$ grain) |
|---|---|---|---|---|
| Manure | | | | |
| Control | 0.893 | 3.218 | 122.3± 2.4 a | 2.8 |
| Compost | 0.951 | 2.491 | 102.6±13.6 ab | 3.8 |
| Cow dung | 0.808 | 2.135 | 79.6±15.6 b | 4.3 |
| LSD $_{0.05}$ | 0.307 | 3.490 | 38.2 | 1.7 |
| Variety | | | | |
| Manawthukha | 1.084 a | 2.585 | 121.5±15.0 a | 3.8 |
| IR-50 | 0.683 b | 2.644 | 81.6±8.0 b | 3.5 |
| LSD $_{0.05}$ | 0.251 | 2.849 | 31.2 | 1.4 |
| Pr>F | | | | |
| Manure | 0.6091 | 0.7922 | 0.0901 | 0.2332 |
| Variety | 0.0046 | 0.9649 | 0.0164 | 0.6719 |
| Manure*Variety | 0.6724 | 0.5766 | 0.6633 | 0.9931 |
| CV (%) | 27.67 | 106.09 | 29.92 | 38.66 |

Within each column, values with different alphabets indicate significant differences among the treatments at 5% of LSD test.

**Nitrous oxide emission.** The seasonal nitrous oxide emissions of rice are shown in Figs 8 (dry season) and 9 (wet season). Relatively high nitrous oxide emissions were found in the early growth stage, and reduced emissions were found in the later growth stage in both seasons. There were significant different in nitrous oxide emissions among the water management practices in dry season (P<0.0159) but not significant in wet season (Table 4). AWD gave a relative high $N_2O$ emission in both seasons. No significant difference was observed in $N_2O$ emission among the manure rates in either season.

**Global warming potential (GWP).** The GWP was significantly different among water management practices in both seasons (P<0.0004, P<0.0000). A higher GWPs (66.3 Mt $CO_2$-eq. $ha^{-1}$ and 40.0 Mt $CO_2$-eq. $ha^{-1}$) were observed under CF than under AWD (20.1 Mt $CO_2$-eq. $ha^{-1}$ and 13.7 Mt $CO_2$-eq. $ha^{-1}$) in the dry (Fig 10A) and wet seasons (Fig 10B), respectively. Generally, the large application of cow dung manure resulted in a higher GWP in both seasons (Fig 12B). The different rates of cow dung manure had no significant effect on the GWP in either season.

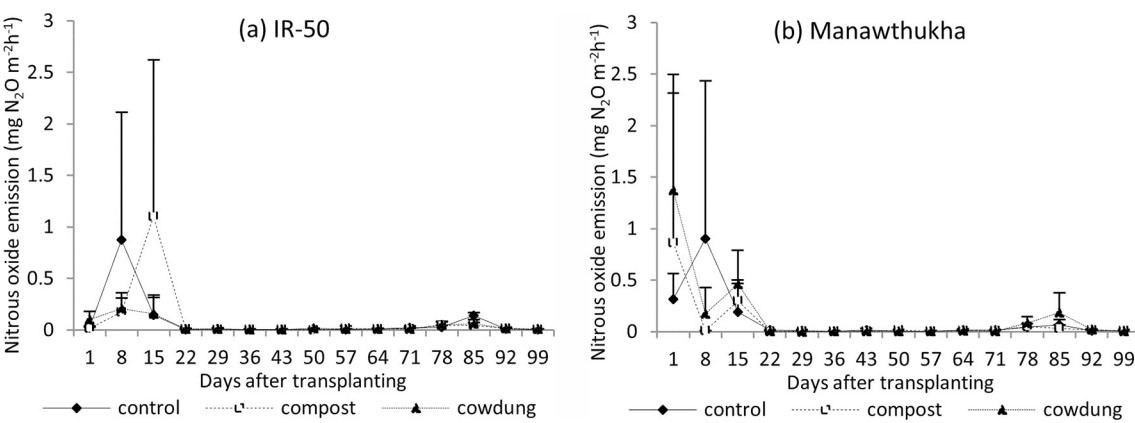

**Fig 4. Nitrous oxide emission of rice varieties at Yezin Agricultural University during the wet season, 2016.** Mean value±standard deviation (n = 3).

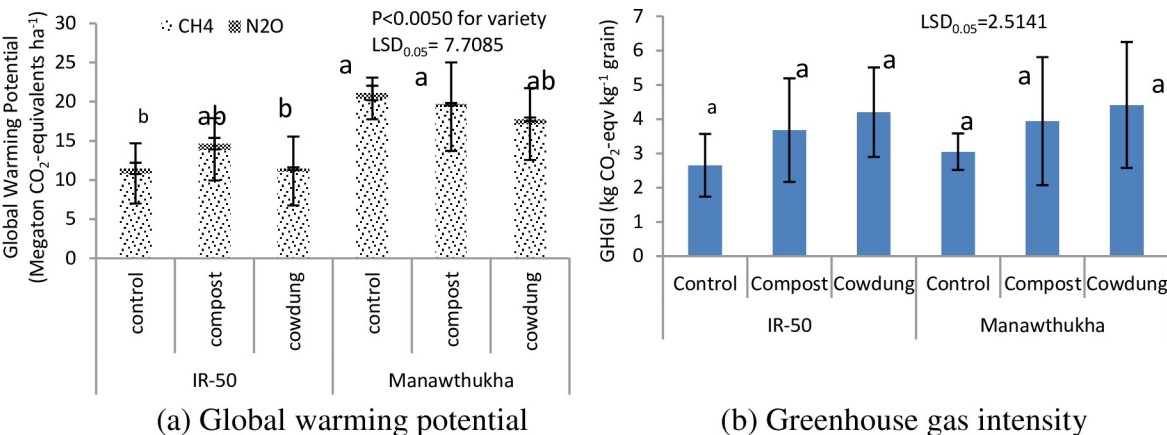

**Fig 5.** Effect of organic manure and rice varieties on (a) global warming potential and (b) greenhouse gas intensity during wet season, 2016. Mean value±standard deviation (n = 3).

**Rice yield.** In this pot experiment, the grain yield was not significantly affected by water management and manure amendment in either rice season (Table 5). However, higher grain yields per plant were recorded with AWD than CF in both seasons. The manure rates had no significant effect on rice yield.

**Greenhouse gas intensity.** Greenhouse gas intensities were significantly different among the water management practices in both seasons (P<0.0002, P<0.0000) (Table 5). Significantly higher GHGI values were found under CF than under AWD in both seasons (Fig 11). No significant differences were found in GHGI values among the manure management practices. However, the higher amount of cow dung manure affected the GHGI values under CF irrigation, but the effect of manure was suppressed by AWD.

**Water input and water saving.** Water inputs of rice as affected by water and manure management are shown in Table 6. There was significantly different of water input among the water management practices in either rice season (P<0.0001, P<0.0000). CF was irrigated more than AWD. The organic manure increases the water holding capacity of the soil. Accordingly, the higher amount of cow dung manure used less water. Water saving of treatments is shown in Table 7. AWD saved water 13.6% over CF in no cow dung manure in dry season and

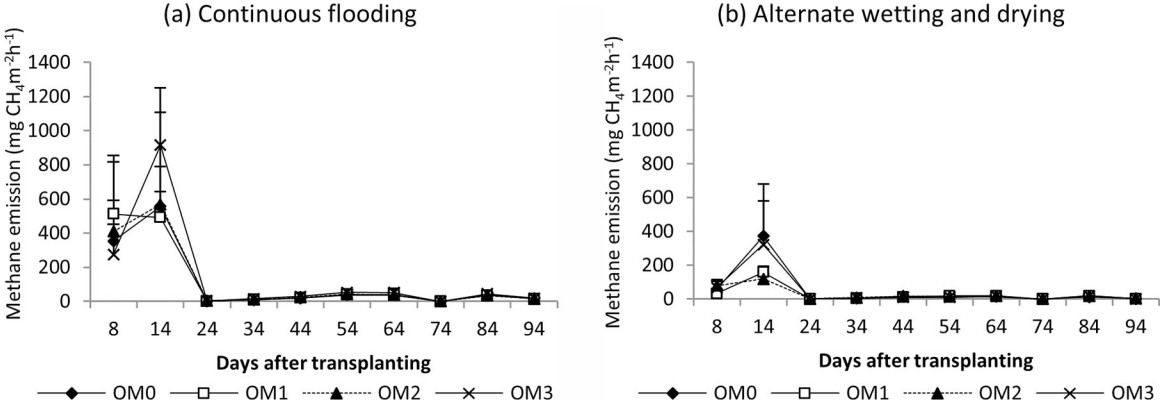

**Fig 6.** Methane emission from rice under (a) continuous flooding and (b) alternate wetting and drying during the dry season, 2017. Mean value±standard deviation (n = 3).

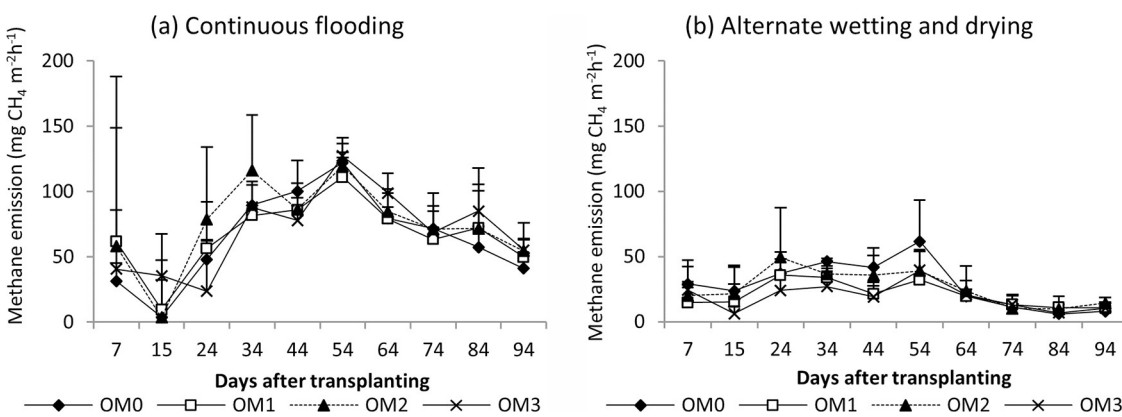

**Fig 7.** Methane emission from rice under (a) continuous flooding and (b) alternate wetting and drying during the wet season, 2017. Mean value±standard deviation (n = 3).

19.1% in wet season. Across the manure management, the water saving of rice mainly depends on water management practices.

## Discussion

### Effect of manure and rice varieties on greenhouse gas emissions, global warming potential and rice yield

Methane production and emissions from flooded paddies are highly affected by the addition of organic matter [47]. The higher $CH_4$ emission in this study was found in later growth stages (Fig 3) because the $CH_4$ emission was associated with higher soil organic matter with increased microbial activities, decomposition of plant residues from fallen leaves and decayed roots, and higher availability of root exudates in the rhizosphere [48]. Although there was no significant

**Table 4. Mean effects of water and manure management on greenhouse gases emission of rice during dry and wet seasons, 2017.**

| Treatment | Cumulative methane emission (g $CH_4$ $kg^{-1}$ soil) | | Cumulative nitrous oxide emission (mg $N_2O$ $kg^{-1}$ soil) | |
|---|---|---|---|---|
| | Dry season | Wet season | Dry season | Wet season |
| Water | | | | |
| CF | 3.327 a | 2.009 a | 0.590 b | 0.183 |
| AWD | 0.996 b | 0.682 b | 1.392 a | 0.344 |
| LSD $_{0.05}$ | 1.089 | 0.369 | 0.631 | 0.176 |
| Manure | | | | |
| $OM_0$ (0 t $ha^{-1}$) | 2.129 | 1.361 | 0.786 | 0.306 |
| $OM_1$ (2.5 t $ha^{-1}$) | 2.036 | 1.279 | 1.621 | 0.201 |
| $OM_2$ (5 t $ha^{-1}$) | 2.007 | 1.460 | 0.817 | 0.234 |
| $OM_3$ (7.5 t $ha^{-1}$) | 2.475 | 1.283 | 0.740 | 0.312 |
| LSD $_{0.05}$ | 1.540 | 0.522 | 0.892 | 0.248 |
| Pr>F | | | | |
| Water | 0.0003 | 0.0000 | 0.0159 | 0.0709 |
| Manure | 0.9114 | 0.8682 | 0.1546 | 0.7335 |
| Water x Manure | 0.9038 | 0.8217 | 0.0890 | 0.8582 |
| CV (%) | 58.21 | 31.74 | 73.53 | 77.08 |

Within each column, values with different alphabets indicate significant differences among the treatments at 5% of LSD test.

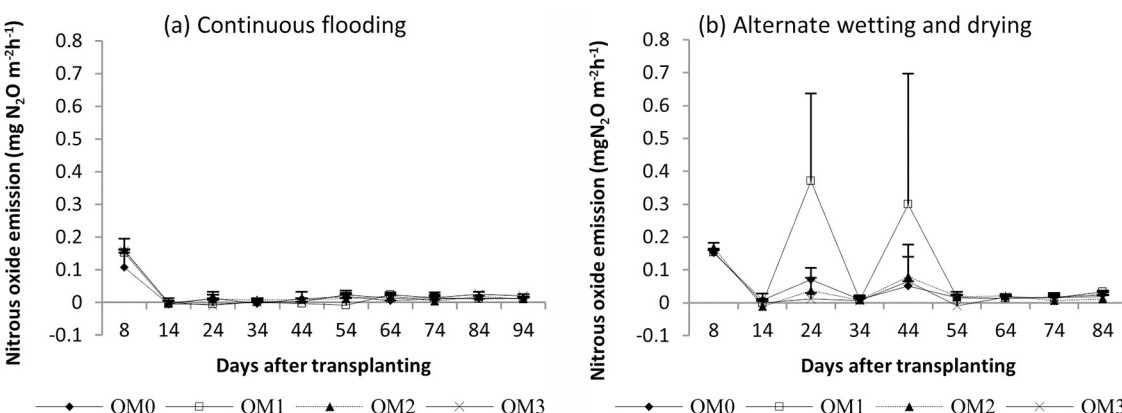

**Fig 8.** Nitrous oxide emission from rice under (a) continuous flooding and (b) alternate wetting and drying during the dry season, 2017. Mean value±standard deviation (n = 3).

difference based on manure management, cow dung manure reduced methane emissions. This study also agrees with that of Oo et al. [30], who reported the lower $CH_4$ emissions from pots treated with well-decomposed cattle manure due to fewer carbon substrates with a reduction of potential $CH_4$ precursors resulting from the preceding decompostion. In contrast to $CH_4$ emissions, high $N_2O$ emissions were recorded in the early growth stage, and small amounts of emissions were recorded in the later growth stage (Fig 4). This could be due to the rapid nitrification with the presence of oxygen and denitrification with the utilization of $NO_3^-$ as electron acceptor in the initial stage with high temperature and low rainfall (Fig 1), and indigenous soil nitrogen (Table 1). After emission peaked initially, the rates of $N_2O$ emission were generally low due to continuous flooding. Another study reported that the consistently low soil redox potential under continuous flooding resulted in more complete denitrification and thus reduced $N_2O$ emissions [49]. The control (no manure) treatment resulted in higher $N_2O$ emissions than the compost and cow dung treatments. Lower nitrous oxide emission was resulted by incorporation of organic inputs due to nitrogen immobilization [50–52]. This study agrees with that of Shan and Yan [53], who reported that $N_2O$ emissions were significantly reduced by crop residue return combined with synthetic N fertilizers compared with emissions from treatments only received synthetic N fertilizer.

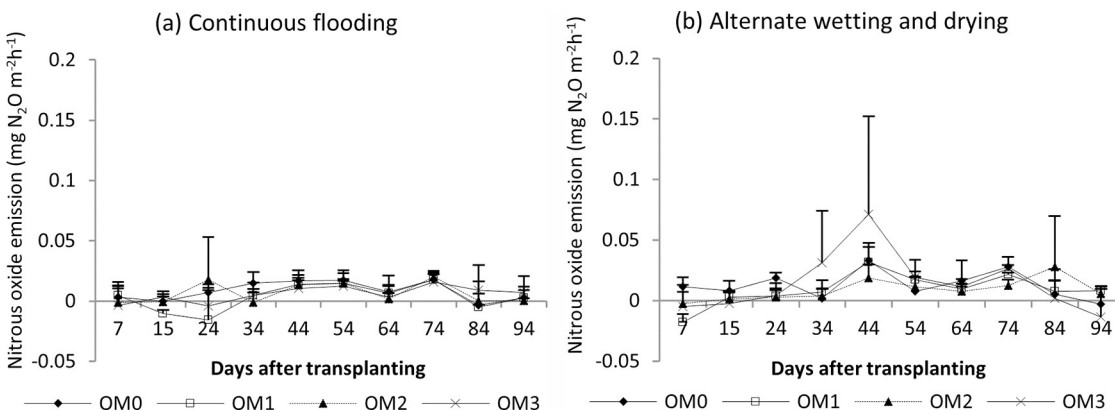

**Fig 9.** Nitrous oxide emission of rice under (a) continuous flooding and (b) alternate wetting and drying during the wet season, 2017. Mean value±standard deviation (n = 3).

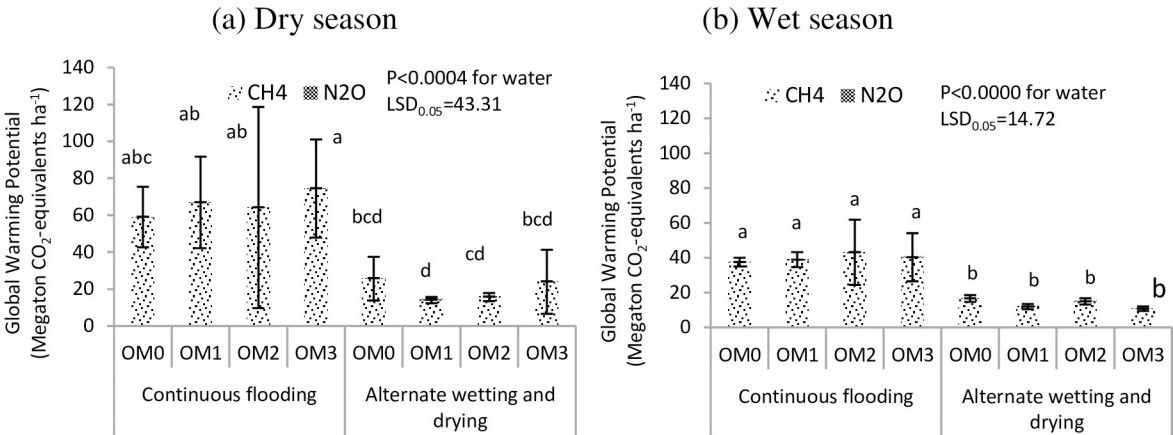

**Fig 10.** Effect of water and cow dung manure management on global warming potential of potted rice during dry (a) and wet (b) seasons, 2017. $OM_0$-no cow dung, $OM_1$- cow dung 2.5 t ha$^{-1}$, $OM_2$- cow dung 5.0 t ha$^{-1}$, $OM_3$- cow dung 7.5 t ha$^{-1}$. Mean value±standard deviation (n = 3).

The selection of suitable rice varieties might play a significant role in regulating CH$_4$ emissions from rice fields [54]. The results of this study showed that there was a significant difference between the tested rice varieties (Fig 3 and Table 3). The result was in agreement with other findings, which highlighted that there were substantial differences in the rates of CH$_4$ emission among different rice varieties [30, 55, 56]. As ninety percent of methane emissions to the atmosphere are through rice plants [57], the Manawthukha variety, which has a longer growth duration, emitted more CH$_4$ than IR-50, which has a shorter growth duration. Previous studies [42, 58–60] have also reported that the CH$_4$ flux from late maturing rice is higher than that from early maturing rice. According to this research finding, a shorter growth duration (IR-50) variety can be used to reduce methane emissions. When compared to the two rice

**Table 5. Mean effects of water and manure management on rice yield during dry and wet seasons, 2017.**

| Treatment | Yield (g plant$^{-1}$) | | GHGI (kg CO$_2$-eq. kg$^{-1}$ grain) | |
|---|---|---|---|---|
| | Dry season | Wet season | Dry season | Wet season |
| Water | | | | |
| CF | 175.06±6.3 | 180.88±4.9 | 7.4 a | 4.3 a |
| AWD | 177.73±6.8 | 183.47±6.7 | 2.1 b | 1.4 b |
| LSD $_{0.05}$ | 12.67 | 14.27 | 2.3 | 0.7 |
| Manure | | | | |
| OM$_0$ (0 t ha$^{-1}$) | 181.54±9.4 | 183.70±3.1 | 4.5 | 2.9 |
| OM$_1$ (2.5 t ha$^{-1}$) | 171.79±3.2 | 178.81±0.1 | 4.7 | 2.7 |
| OM$_2$ (5 t ha$^{-1}$) | 178.08±11.0 | 179.20±0.1 | 4.3 | 3.1 |
| OM$_3$ (7.5 t ha$^{-1}$) | 174.16±5.0 | 187.01±7.6 | 5.6 | 2.7 |
| LSD $_{0.05}$ | 17.92 | 20.18 | 3.2 | 1.1 |
| Pr>F | | | | |
| Water | 0.6608 | 0.7060 | 0.0002 | 0.0000 |
| Manure | 0.6758 | 0.7984 | 0.8359 | 0.8283 |
| Water x Manure | 0.3976 | 0.6511 | 0.7742 | 0.8092 |
| CV (%) | 8.30 | 9.05 | 55.13 | 31.13 |

Within each column, values with different alphabets indicate significant differences among the treatments at 5% of LSD test.

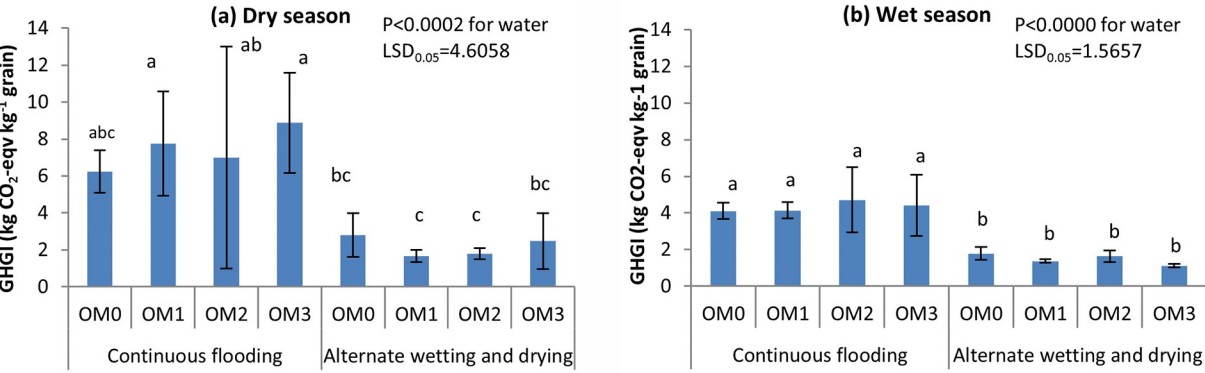

**Fig 11.** Effect of water and cow dung manure management on greenhouse gas intensity of potted rice during dry (a) and wet (b) seasons, 2017. $OM_0$-no cow dung, $OM_1$- cow dung 2.5 t ha$^{-1}$, $OM_2$- cow dung 5.0 t ha$^{-1}$, $OM_3$- cow dung 7.5 t ha$^{-1}$. Mean value±standard deviation (n = 3).

varieties, IR-50 resulted in higher $N_2O$ emissions than the Manawthukha variety. This could be due to the favourable effect of root exudates of IR-50 on the nitrification process in the soil. Gogoi and Baruah [25] reported that the main driving forces influencing $N_2O$ emission in rice were soil $NO_3$-N, soil organic carbon. There was no interaction between manure and rice varieties. The compost and Manawthukha variety emitted higher $CH_4$ due to longer decomposing time and support of substrates for methanogens. The control treatment and IR-50 variety resulted in higher $N_2O$ emission due to nitrification and denitrification of inorganic nutrient in the favour of root exudates (Fig 5A).

Cow dung manure resulted in a 10.4% reduction in GWP compared to the control (no manure) with reduced $CH_4$ and $N_2O$ emission. Combination of decomposed cow dung manure + mineral fertilizer might suppress the available carbon and nitrogen for $CH_4$ and $N_2O$ production. Unfortunately, in our study, we couldn't measure the carbon and nitrogen at every gas sampling and their mechanisms. While an increased GWP (5.5%) was found in the compost compared to the control. Huang et al. [61] reported that incorporation of organic residues provides a source of readily available C and N in the soil and subsequently influences $N_2O$ emissions. Manawthukha had a higher GWP (56%) than IR-50. These results were supported by Zheng et al. [62], who reported that yield-scaled GWP at 80–90 days of growth duration after transplanting was 87% higher than that at 70–80 days of growth duration after

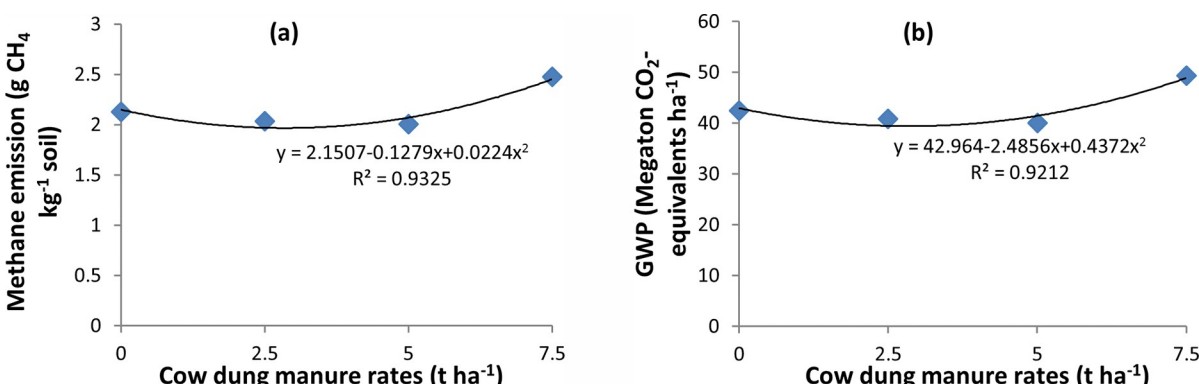

**Fig 12.** Relationship between methane emission and different cow dung manure rates (a) and between GWP and different cow dung manure rates (b) in dry season, 2017.

**Table 6. Mean effects of water and manure management on water input of rice during dry and wet seasons, 2017.**

| Treatment | Water input $_{(I+R)}$ (mm) | |
|---|---|---|
| | **Dry season** | **Wet season** |
| Water | | |
| CF | 1075.5 a | 842.1 a |
| AWD | 969.8 b | 681.4 b |
| LSD $_{0.05}$ | 42.4 | 32.7 |
| Manure | | |
| $OM_0$ | 1030.5 | 775.0 |
| $OM_1$ | 1017.6 | 750.5 |
| $OM_2$ | 1015.4 | 740.6 |
| $OM_3$ | 1027.1 | 780.9 |
| LSD $_{0.05}$ | 60.0 | 46.2 |
| Pr>F | | |
| Water | 0.0001 | 0.0000 |
| Manure | 0.9385 | 0.2372 |
| Water x Manure | 0.0955 | 0.7619 |
| CV (%) | 4.80 | 4.96 |

Within each column, values with different alphabets indicate significant differences among the treatments at 5% of LSD test.

transplanting. Feng et al. [63] also reported that yield-scaled GWP in late rice in a double-rice cropping system was 73% higher than that for varieties with 90–100 days of growth duration after transplanting.

A high grain yield of rice was recorded from the control due to the mineralization of indigenous soil organic matter and inorganic fertilizer (Table 3). Oo et al. [64] reported a high grain yield under inorganic sources of nutrients due to the immediate release and availability of nutrients. Relative to IR-50, Manawthukha had a higher grain yield by 48.9% whereas the GWP was reduced in IR-50 than in Manawthukha. Therefore, considering the mitigation practice for GWP from rice production, IR-50 and cow dung manure was experimented in 2017 rice seasons.

## Effect of water management and manure application on greenhouse gas emissions, global warming potential and rice yield

In the dry season, high methane emissions were observed in the early growth stage and then decreased until harvest in both water regimes (Fig 6). The early increase in $CH_4$ emission was

**Table 7. Comparison of water saving of rice as affected by water and cow dung manure management during dry and wet seasons, 2017.**

| Treatment | | Water saving (%) | |
|---|---|---|---|
| **Water** | **Cow dung manure** | **Dry season** | **Wet season** |
| CF | $OM_0$ | | |
| CF | $OM_1$ | | |
| CF | $OM_2$ | | |
| CF | $OM_3$ | | |
| AWD | $OM_0$ | 13.6 | 19.1 |
| AWD | $OM_1$ | 7.5 | 18.7 |
| AWD | $OM_2$ | 2.8 | 17.1 |
| AWD | $OM_3$ | 14.9 | 21.2 |

due to the indigenous soil carbon content and availability of substrates, and the decrease in the later growth stage was due to the senescence of older leaves and non-availability of substrate as the crop approached maturity [65–67]. In the wet season, a slight increase in emission was recorded in the early growth stage, and emission was highest in the middle stage and gradually decreased until harvest (Fig 7). A higher rate of $CH_4$ production is attributed to the availability of organic substrates from the previous crop residues in the form of plant-derived C through processes such as root exudation and release of fallen leaves and intensively reduced conditions in the rice rhizosphere [54, 68]. The gradual decrease in $CH_4$ emission in later growth stage was due to the decomposition and non-availability of substrates depending on temperature (Fig 2). Higher cumulative $CH_4$ emissions were observed under CF than under AWD in both seasons because the anoxic conditions increased the methanogen population and favoured methane production. The methane emission was higher in the dry season than in the wet season because it depended on the availability of indigenous soil carbon and decomposition process favoured by high temperature in the dry season, and depletion of soil carbon and previous crop resides decomposition hindered by low temperature in wet season (Fig 2). Relatively high nitrous oxide emissions were observed in the early growth stage, and reduced emissions were observed in the later growth stage in both seasons (Figs 8 and 9). This was due to the indigenous soil nitrogen content in the early stage, and depletion of nutrients and the available nitrogen content for nitrification and denitrification in the later stage. Higher $N_2O$ emissions were observed under AWD than CF in both seasons [54, 69, 70]. AWD increased $N_2O$ emissions by 135% compared to CF in dry season and 88% in wet season. Similar to the $CH_4$ emission pattern, higher $N_2O$ emissions were observed in the dry season than in the wet season because of the soil nitrogen availability and the favoured soil condition (high temperature and low rainfall) for nitrification and denitrification in the dry season [71, 72].

Generally higher rates of cow dung manure resulted in higher $CH_4$ and $N_2O$ emission because it provided carbon and nitrogen sources for methanogenesis and nitrification and denitrification process. Methane is produced by methanogenic bacteria during the anaerobic digestion of organic substrates [4] and $N_2O$ production is observed by soil water content and availability of substrates (nitrate and easily degradable organic matter) for denitrification [73]. However, the polynomial distribution of methane emission with different cow dung manure rates (Fig 12A) and polynomial distribution of GWP with different cow dung manure rates (Fig 12B) were observed in dry season. The polynomial regression equation for methane emission is $y = 2.1507–0.1279x+0.0224x^2$. The equation showed that the methane emission decreased in the rate of 0.1279 g $CH_4$ $kg^{-1}$ soil at every increased unit (ton) of cow dung manure applied, after that increased in the rate of 0.0224 g $CH_4$ $kg^{-1}$ soil at every increased unit (ton) of cow dung manure applied. The coefficient of determination ($R^2$) showed that 93% of variation in methane emission could be accounted by the quadratic regression equation of different cow dung manure rates. The polynomial regression equation for GWP is $y = 42.964–2.4856x+0.4372x^2$. The equation showed that the GWP decreased in the rate of 2.4856 Mt $CO_2$-equivalent $ha^{-1}$ at every increased unit (ton) of cow dung manure applied, after that increased in the rate of 0.4372 Mt $CO_2$-equivalent $ha^{-1}$ at every increased unit (ton) of cow dung manure applied. The coefficient of determination ($R^2$) showed that 92% of variation in GWP could be accounted by the quadratic regression equation of different cow dung manure rates. According to these results, it would be recommended that 5 t $ha^{-1}$ should be applied for improving soil fertility and reduced greenhouse gas emission. No interactive effect between water management and the application of different cow dung manure rates was found on $CH_4$ emissions in either season. However, the effect of the application of different cow dung manure rates on $N_2O$ emissions was influenced by alternate wetting and drying irrigation

practices in the dry season but not in the wet season. Therefore, the nitrification and denitrification processes of soil are influenced by the soil moisture content [74–77].

A higher GWP was observed under CF in accordance with the higher methane emissions in both seasons (Fig 10). Methane emission mainly contributes to the GWP from paddy production. Many studies [39, 54, 78–82] reported that $N_2O$ emissions contribute much less to the global warming potential than those of $CH_4$. Therefore, the water regime in paddy production is the main factor controlling $CH_4$ emissions from rice fields [39, 54, 83].

AWD resulted in a higher grain yield per plant than did CF in both seasons (Table 3) because it strengthened the air exchange between the soil and atmosphere and supplied sufficient oxygen to the root system to accelerate soil organic matter mineralization which increase soil fertility and favour rice growth [84, 85]. Yang and Zhang [86] reported an increase in paddy yield under AWD due to the increase in the proportion of productive tillers. In this study, AWD saved water about 10% over CF in dry season and 19% in wet season. Zhang et al. [87] also indicated water saving of 35% under AWD with a 10% yield increase relative to that under CF. Liu et al. [88], Ye et al. [89], and Djaman et al. [90] found that grain yield increased with reduced water input by AWD. Grain yield was not affected by the different rates of cow dung manure application because its decomposition was influenced by biotic and soil-environmental factors. Relative to the CF water management, AWD produced comparable grain yields, increased by 1.5% and decreased GHGI by 69% (Table 5). This suggests that by adopting alternate wetting and drying irrigation, it would be possible to achieve the dual goals of maintaining productivity while minimizing the global warming potential of rice cultivation.

## Conclusion

For sustainable agriculture, organic manures should be added at a recommended amount to improve the rice yield and reduce greenhouse gas emissions. From our findings, the application of cow dung manure can be recommended in paddy production since it mitigated the global warming potential compared to that of the control and compost groups, although it resulted in a lower yield potential. Additionally, the short duration rice variety had a lower GWP and a lower GHGI value while maintaining the potential rice yield. Manure-induced greenhouse gas emissions were suppressed by AWD irrigation practices, with significant GHGI values. Thus, short duration varieties are highly recommended with AWD irrigation and application of 5 t ha$^{-1}$ cow dung manure to reduce greenhouse gas emissions and maintain rice yield under the soil-environmental conditions of Myanmar. In our study, there was no relationship between GWP and rice yield. Therefore, the choice of rice varieties should be combined with soil-environmental factors and cultivation systems to mitigate greenhouse gas emissions while increasing rice yields for sustainable rice production. Further studies under field condition are needed to measure the effect of manure and mineral fertilizer on greenhouse gas emission, global warming potential and rice yield under water management conditions for better understanding of emission mechanisms.

## Acknowledgments

We deeply thank Intergovernmental Panel on Climate Change (IPCC) for research grant. We also thank U Thein Myint Tun (Staff Officer of Department of Agriculture—retired) for his support to the researches and Dr. Yoshinori YAMAMOTO (Professor, Emeritus, Faculty of Agriculture, Kochi University, Japan) and Dr. Koki TOYOTA (Professor, Graduate School of Bio-Applications and Systems Engineering, Tokyo University of Agriculture and Technology, Japan) for their technical support to our study.

## Author Contributions

**Project administration:** Kyaw Kyaw Win.

**Writing – original draft:** Ei Phyu Win.

**Writing – review & editing:** Sonoko D. Bellingrath-Kimura, Aung Zaw Oo.

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
