## [Decision Letter · Decision Letter 0]

16 Dec 2020

PONE-D-20-31281

Influence of rice varieties, organic manure and water management on greenhouse gas emissions from paddy rice soils

PLOS ONE

Dear Dr. Win,

Thank you for submitting your manuscript to PLOS ONE. After careful consideration, we feel that it has merit but does not fully meet PLOS ONE’s publication criteria as it currently stands. Therefore, we invite you to submit a revised version of the manuscript that addresses the points raised during the review process.

Although the reviewer #1 recommended to reject this manuscript, I believe this manuscript deserves another chance to improve. So, please make sure to address all concerns raised by all three reviewers. I look forward to reading your revised manuscript.

We look forward to receiving your revised manuscript.

Kind regards,

Debjani Sihi

Academic Editor

PLOS ONE

Journal Requirements:

2.  Thank you for submitting the above manuscript to PLOS ONE. During our internal evaluation of the manuscript, we found significant text overlap between your submission and the following previously published works, some of which you are an author.

http://www.jeb.co.in/journal_issues/201609_sep16/paper_24.pdf

https://scialert.net/fulltext/?doi=ajps.2010.414.422

https://www.mdpi.com/2073-4441/10/6/711/html

Please revise the manuscript to rephrase the duplicated text, cite your sources, and provide details as to how the current manuscript advances on previous work. Please note that further consideration is dependent on the submission of a manuscript that addresses these concerns about the overlap in text with published work.

Reviewers' comments:

Reviewer's Responses to Questions

**Comments to the Author**

1. Is the manuscript technically sound, and do the data support the conclusions?

Reviewer #1: Yes

Reviewer #2: Partly

Reviewer #3: Partly

2. Has the statistical analysis been performed appropriately and rigorously? 

Reviewer #1: Yes

Reviewer #2: Yes

Reviewer #3: No

3. Have the authors made all data underlying the findings in their manuscript fully available?

Reviewer #1: Yes

Reviewer #2: Yes

Reviewer #3: Yes

4. Is the manuscript presented in an intelligible fashion and written in standard English?

Reviewer #1: Yes

Reviewer #2: No

Reviewer #3: Yes

5. Review Comments to the Author

Reviewer #1: Some statistical analysis information is missing especially in the tables but this can be corrected if the authors have data. Also more information on the analysis methods have been requested in the attached reviewer notes.

Reviewer #2: Comments to the author and editors

The study is focused on impact of manure application, rice verities and water management on GHG emission from paddy soil putted in pot experiment. The objectives of this study were a) to assess the effect of different types of manure amendments and rice varieties on greenhouse gas emissions and b) to determine the optimum manure application rate to increase rice yield while mitigating GHG emissions under alternate wetting and drying irrigation in paddy rice production. I appreciate the effort of the worker for reporting data, however, the study has been conducted in the pot and data has reported for only one season which is not adequate to draw sound conclusion. It would have been more appropriate if at least two year data might have been generated. There are many research works already reported on this aspect with long duration of experiment from field trial. The manuscript also lack proper synthesis and sound discussion. It may be suitable for publication in local journal of the study area. The some comments has been made in the pdf of the manuscript and some broad comments are enlisted below

1. The readability of the manuscript is poor

2. The abstract lack genesis, objective and conclusion of the research

3. Material and methods lack: information on feed-stock of compost, clear statement regarding control and N-fertilization, intercultural operation, harvesting date, dimension of gas chamber and duration of gas sampling. Cumulative CH4 emission is highly dependent on duration of gas sampling and biomass yield especially number of tiller. No data has been presented on vegetative growth of the two rice varieties.

4. The clear difference in compost and cow dung manure has not been stated: the method of preparation, feed-stock used etc.

5. Table 2.1 and 2.2 should be similar in terms of parameters reported

6. The result section and discussion need re-synthesis to draw final conclusion. Control has shown lowest GHG intensity in first experiment, but Cow dung application has been recommended. Need proper discussion

7. In second experiment, increasing rate of cow dung has reduced CH4 emission as compared to control (control> C 2.5 t/ha > C 5 t/ha < c 7.5 t/ha) up to 5 t/ha then it increased. Need detailed discussion on this trend

Reviewer #3: The statistical design used in the not suitable for this study. The authors have to use CRD design for analysing their data. Split plot design used by authors is not for pot experiments. This type of design is used for field experiments. This is why I feed the the manuscript is not technically sound.

I suggest the authors to revise the results based on revised statistical analysis and modified reporting of pot experiment data as pot experiment result are not reported on per hectare basis

Not able to add orchid id due to verification issues

6. PLOS authors have the option to publish the peer review history of their article (what does this mean?). If published, this will include your full peer review and any attached files.

Reviewer #1: No

Reviewer #2: No

Reviewer #3: No

---

## [Author Response · Author response to Decision Letter 0]

11 Mar 2021

Dear Sir,

I really thank you very much for your valuable and constructive comments on this manuscript. I have revised all reviewers’ comments and may I know if you have more comments and suggestions to be manuscript improved.

Reviewer #1: Some statistical analysis information is missing especially in the tables but this can be corrected if the authors have data. Also more information on the analysis methods have been requested in the attached reviewer notes.

Answer: I have revised them.

Reviewer #2: Comments to the author and editors

The study is focused on impact of manure application, rice verities and water management on GHG emission from paddy soil putted in pot experiment. The objectives of this study were a) to assess the effect of different types of manure amendments and rice varieties on greenhouse gas emissions and b) to determine the optimum manure application rate to increase rice yield while mitigating GHG emissions under alternate wetting and drying irrigation in paddy rice production. I appreciate the effort of the worker for reporting data, however, the study has been conducted in the pot and data has reported for only one season which is not adequate to draw sound conclusion. It would have been more appropriate if at least two year data might have been generated. There are many research works already reported on this aspect with long duration of experiment from field trial. The manuscript also lack proper synthesis and sound discussion. It may be suitable for publication in local journal of the study area. The some comments has been made in the pdf of the manuscript and some broad comments are enlisted below

1. The readability of the manuscript is poor

2. The abstract lack genesis, objective and conclusion of the research

3. Material and methods lack: information on feed-stock of compost, clear statement regarding control and N-fertilization, intercultural operation, harvesting date, dimension of gas chamber and duration of gas sampling. Cumulative CH4 emission is highly dependent on duration of gas sampling and biomass yield especially number of tiller. No data has been presented on vegetative growth of the two rice varieties.

4. The clear difference in compost and cow dung manure has not been stated: the method of preparation, feed-stock used etc.

5. Table 2.1 and 2.2 should be similar in terms of parameters reported

6. The result section and discussion need re-synthesis to draw final conclusion. Control has shown lowest GHG intensity in first experiment, but Cow dung application has been recommended. Need proper discussion

7. In second experiment, increasing rate of cow dung has reduced CH4 emission as compared to control (control> C 2.5 t/ha > C 5 t/ha < c 7.5 t/ha) up to 5 t/ha then it increased. Need detailed discussion on this trend

Answer: Sir, thank you very much for your constructive comments and valuable usage of words. I had revised the abstract with your comments and suggestions and I substituted your comments to our manuscript to be improved of readability of it. I have revised all comments. Please may I know more comments and suggestions to be improved of our manuscript and I really thank you and appreciate your valuable comments.

Reviewer #3: The statistical design used in the not suitable for this study. The authors have to use CRD design for analysing their data. Split plot design used by authors is not for pot experiments. This type of design is used for field experiments. This is why I feed the the manuscript is not technically sound.

I suggest the authors to revise the results based on revised statistical analysis and modified reporting of pot experiment data as pot experiment result are not reported on per hectare basis

Answer: Sir, I revised the manuscript with your suggestion. I revised the statistical analysis to two factors with CRD design. Thank you very much for your constructive and valuable comments. Please may I know more comments and suggestions and I really appreciate your fruitful comments.

---

## [Decision Letter · Decision Letter 1]

26 May 2021

PONE-D-20-31281R1

Influence of rice varieties, organic manure and water management on greenhouse gas emissions from paddy rice soils

PLOS ONE

Dear Dr. Ei,

Thank you for submitting your manuscript to PLOS ONE. After careful consideration, we feel that it has merit but does not fully meet PLOS ONE’s publication criteria as it currently stands. Therefore, we invite you to submit a revised version of the manuscript that addresses the points raised during the review process.

Reviewer #3 is still not convinced with the modifications in the revised manuscripts. I will encourage you to address all concerns raised my reviewer #3. 

We look forward to receiving your revised manuscript.

Kind regards,

Debjani Sihi

Academic Editor

PLOS ONE

Reviewers' comments:

Reviewer's Responses to Questions

**Comments to the Author**

1. If the authors have adequately addressed your comments raised in a previous round of review and you feel that this manuscript is now acceptable for publication, you may indicate that here to bypass the “Comments to the Author” section, enter your conflict of interest statement in the “Confidential to Editor” section, and submit your "Accept" recommendation.

Reviewer #2: (No Response)

Reviewer #3: (No Response)

2. Is the manuscript technically sound, and do the data support the conclusions?

Reviewer #2: (No Response)

Reviewer #3: Partly

3. Has the statistical analysis been performed appropriately and rigorously? 

Reviewer #2: (No Response)

Reviewer #3: Yes

4. Have the authors made all data underlying the findings in their manuscript fully available?

Reviewer #2: (No Response)

Reviewer #3: Yes

5. Is the manuscript presented in an intelligible fashion and written in standard English?

Reviewer #2: (No Response)

Reviewer #3: Yes

6. Review Comments to the Author

Reviewer #2: (No Response)

Reviewer #3: The authors have modified the manuscript to a larger extent. But there are several points which are not acceptable. The authors say the emission of methane in we season is less compared to dry season.

The addition of carbon due to application of 2.5,5 and 7.5 t of manure having 16% C will be quite different. In fact the carbon will be 2 and 3 times more in quantitative terms in OM2 and OM3. (Addition of 2.5 t of manure will add 400 kg of carbon, while OM2 and OM3 will add 800kg and 1200 kg of carbon. It is very surprising that addition of such high amount of carbon is not increasing methane emission even in wet season. It seems there is serious sampling issue. There may be some leakage issue either from the chamber or after sampling or error during analysis .

Another point is at one place authors are saying manure quantity was according to N required but they adjusted to maintain same C content, not clear very mind boggling

They say High methane emission and low N2O emission from long duration variety However IR 50 is short duration but N2O is high. The justification given is not very appealing.

There are several other points which are given in the PDF.

7. PLOS authors have the option to publish the peer review history of their article (what does this mean?). If published, this will include your full peer review and any attached files.

Reviewer #2: **Yes: **DIPAK KUMAR GUPTA

Reviewer #3: No

---

## [Author Response · Author response to Decision Letter 1]

28 May 2021

Dear Sir,

I really thank you very much for your valuable and constructive comments and time sharing on this manuscript ‘Influence of rice varieties, organic manure and water management on greenhouse gas emissions from paddy rice soils’. We sincerely appreciate valuable and positive comments from you. We have taken all the comments and suggestions into account in the revised manuscript. 

Sir, I really sorry for your time and deeply thank for that. Your comments made me improved for my profession and I got invaluable knowledge and fruitful remarks. Please may I know if you have more comments and suggestions to be manuscript improved.

Sincerely yours,

Reviewer #3: 

Comment 1

Q: The authors have modified the manuscript to a larger extent. But there are several points which are not acceptable. The authors say the emission of methane in wet season is less compared to dry season. 

A: Sir, these data are realistic. So I have to interpret the results according to research findings. Sir, I conducted the field experiment simultaneously with this pot experiment. In that field experiment, the emission is more in wet season than in dry season but in this pot experiment the reverse is found. Therefore, for this pot experiment, I interpreted the emission is influenced by control environmental factors and mainly depends on the soil nutritional status. (Line: 560-564)

Comment 2

Q: The addition of carbon due to application of 2.5,5 and 7.5 t of manure having 16% C will be quite different. In fact the carbon will be 2 and 3 times more in quantitative terms in OM2 and OM3. (Addition of 2.5 t of manure will add 400 kg of carbon, while OM2 and OM3 will add 800kg and 1200 kg of carbon. It is very surprising that addition of such high amount of carbon is not increasing methane emission even in wet season. It seems there is serious sampling issue. There may be some leakage issue either from the chamber or after sampling or error during analysis. 

A: Sir, thank you very much for your constructive comment and it is improved to my knowledge. Yes, I was also frustrated about this finding. Therefore, according to my knowledge, I interpreted the water management is key player to emission mechanism especially CH4 emission nevertheless of how much addition of cowdung manure. Sir, I also agree with your suggestion about sampling issue on this point. 

Comment 3

Q: Another point is at one place authors are saying manure quantity was according to N required but they adjusted to maintain same C content, not clear very mind boggling.

A: Sir, as pilot experiment, we would just like to know the effect of manures on emission. At that time, we considered that we will replace the chemical fertilizer with manures. And then, we also wanted to compare the effect of manures on emission. Therefore, we considered on same amount of Carbon of manures to get the equal chance of carbon input.

Comment 4

Q: They say High methane emission and low N2O emission from long duration variety However IR 50 is short duration but N2O is high. The justification given is not very appealing.

A: Sir, high CH4 emission and low N2O emission from long duration variety, Manawthukha.

 Low CH4 emission and high N2O emission from short duration variety, IR 50. 

Although CH4 emission depends on longer growth duration for methanogenesis, N2O emission depends on nutritional features of the soil for nitrification and denitrification process. Please may I know for your suggestion upon this response. 

There are several other points which are given in the PDF.

---

## [Editor Report · Decision Letter 2]

14 Jun 2021

Influence of rice varieties, organic manure and water management on greenhouse gas emissions from paddy rice soils

PONE-D-20-31281R2

Dear Dr. Win,

We’re pleased to inform you that your manuscript has been judged scientifically suitable for publication and will be formally accepted for publication once it meets all outstanding technical requirements.

Kind regards,

Debjani Sihi

Academic Editor

PLOS ONE
---

## [Editor Report · Acceptance letter]

21 Jun 2021

PONE-D-20-31281R2 

Influence of rice varieties, organic manure and water management on greenhouse gas emissions from paddy rice soils 

Dear Dr. Win:

I'm pleased to inform you that your manuscript has been deemed suitable for publication in PLOS ONE. Congratulations! Your manuscript is now with our production department. 

Kind regards, 

on behalf of

Dr. Debjani Sihi 

Academic Editor

PLOS ONE